# Tracking DOT1L methyltransferase activity by stable isotope labelling using a selective synthetic co-factor
Nicole Trainor [1,5], Harry J. Whitwell[2,3,5], Beatriz Jiménez[2,3], Katie Addison[1], Emily Leonidou[1,4], Peter A. DiMaggio[4] & Matthew J. Fuchter [1] ✉

Epigenetic processes influence health and disease through mechanisms which alter gene expression. In contrast to genetic changes which affect DNA sequences, epigenetic marks include DNA base modifications or post-translational modification (PTM) of proteins. Histone methylation is a prominent and versatile example of an epigenetic marker: gene expression or silencing is dependent on the location and extent of the methylation. Protein methyltransferases exhibit functional redundancy and broad preferences for multiple histone residues, which presents a challenge for the study of their individual activities. We developed an isotopically labelled analogue of co-factor S-adenosyl-L-methionine ($^{13}CD_3$-BrSAM), with selectivity for the histone lysine methyltransferase DOT1L, permitting tracking of methylation activity by mass spectrometry (MS). This concept could be applied to other methyltransferases, linking PTM discovery to enzymatic mediators.

Post-translational methylation of proteins is a vital cellular process most broadly recognised for its role in the epigenetic modulation of gene expression. Methylation of lysine residues in histone proteins is facilitated by histone lysine methyltransferases (HKMTs) using an S-adenosyl-L-methionine (SAM) co-factor [we note that the term co-factor and co-substrate are used interchangeably for SAM in the HKMT literature]. Subsequent activation or repression of gene transcription is dependent on the position of the methylated lysine and the number of methyl groups present, as well as the composition of PTMs on other residues (Fig. 1a)[1]. Extensive non-histone substrates for HKMTs have also been discovered, though the functional consequences of non-histone methylation remain less well understood[2]. Aberrant histone methyltransferase activity is common in cancer, leading to transcriptional mis-regulation. As such, HKMTs are targets for novel cancer therapies[3,4]. Despite this, significant gaps exist in our biological understanding of protein methylation, driven, in part, by incomplete knowledge surrounding the sites and redundancy of methylation throughout the proteome.

Methylation can be mapped throughout complex proteomes using proteomic techniques such as heavy-methyl Stable Isotope Labelling by Amino acids in Cell culture (hmSILAC[5]/ihmSILAC[6]), which is indispensable for the identification and validation of methyl-peptides by mass spectrometry (MS). Without such isotopic labelling, discovery proteomics for methylation can suffer from unacceptably high false discovery rates, largely due to the misidentification of amino acids during database searching and chemical methylation/ethylation during proteomics sample preperation[7]. hmSILAC-MS utilises cells grown in a "heavy" culture medium containing methionine with a stable isotopic label ($^{13}CD_3$-methionine), which is endogenously converted to $^{13}CD_3$-SAM. Subsequently, the heavy methyl group is incorporated onto target protein substrates through the action of methyltransferases. Newly methylated sites can be identified by MS through an isotopic shift relative to pre-existing ($CH_3$) or alternately labelled methyl markers. Thus, the location, extent, and turnover of protein methylation over time can be determined via hmSILAC-MS.

Despite this, one of the limitations of the hmSILAC-MS approach is the absence of robust methods to attribute the enzyme responsible for a given methyl mark. The chemical biology of HKMTs is rendered complex by their functional redundancy; multiple HKMTs can methylate a common histone residue and several HKMTs have multiple histone substrates. For example, G9a is one of six human HKMTs responsible for methylating lysine 9 of histone 3 (H3K9), and is also capable of methylating H3K27[8,9]. Therefore, selective inhibition or genetic knockdown of a single HKMT may not result in discernible changes to the protein methylome or target residue.

[1]Department of Chemistry, Molecular Sciences Research Hub, Imperial College London, White City Campus, 82 Wood Lane, London W12 OBZ, UK. [2]National Phenome Centre and Imperial Clinical Phenotyping Centre, Department of Metabolism, Digestion and Reproduction, IRDB, Building Imperial College London, London W12 ONN, UK. [3]Section of Bioanalytical Chemistry, Division of Systems Medicine, Department of Metabolism, Digestion and Reproduction, Sir Alexander Fleming Building, Imperial College London, London SW7 2AZ, UK. [4]Department of Chemical Engineering, Imperial College London, South Kensington Campus, London SW7 2AZ, UK. [5]These authors contributed equally: Nicole Trainor, Harry J. Whitwell. ✉e-mail: m.fuchter@imperial.ac.uk

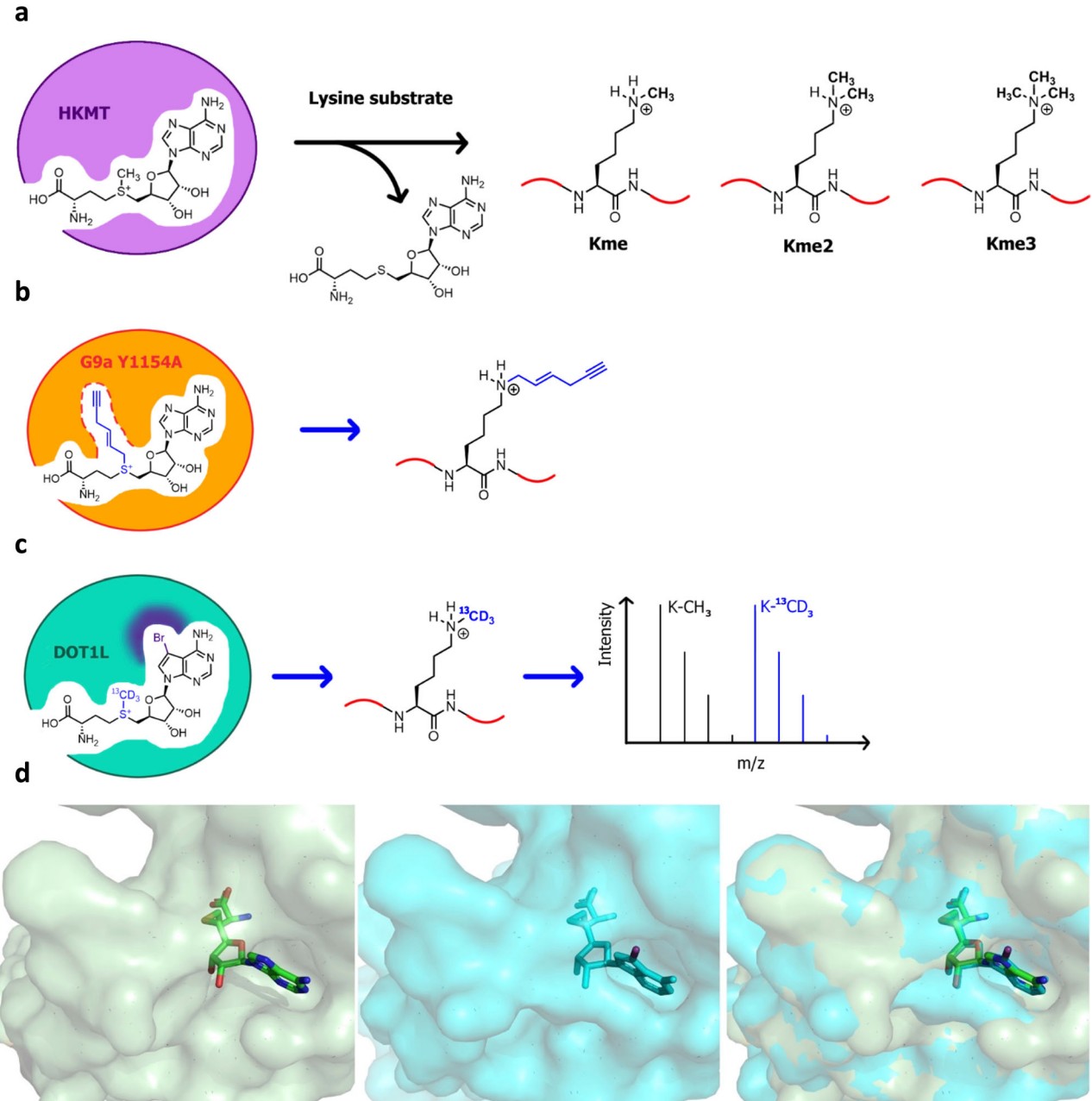

**Fig. 1 | Overview of previous work in the field and the approach developed in this study to track HKMT activity. a** HKMTs catalyse turnover of the SAM co-factor to generate mono-, di- or trimethylated lysine residues on protein substrates; (**b**) application of the bump-and-hole approach to track lysine labelling by the HKMT G9a. A co-factor analogue bearing an alkyl "bump" was designed to selectively bind an engineered form of G9a featuring a "hole" created by a Y1154A mutation; (**c**) MS method of detecting protein methylation by DOT1L developed in this work. The selective co-factor analogue, $^{13}CD_3$-BrSAM, binds to DOT1L to transfer a heavy methyl group to a protein substrate, which is digested to peptides and analysed by MS. The $^{13}CD_3$-labelled peptide appears with a + 4 isotopic shift relative to the corresponding endogenous light methyl peptide; (**d**) X-ray co-crystal structures of SAH:DOT1L (left, green, PDB 3QOX) and bromo-deaza-SAH (**5**):DOT1L (middle, turquoise, PDB 3SX0). An overlay of the two inhibitors is shown on the right, RMSD = 0.092 Å.

Synthetic co-factor analogues have been applied to track the activity of individual methyltransferases, through modification of the active electrophile (i.e., a non-methyl group)[10]. For example, Al Temimi et al. have demonstrated alkylation of lysine-containing peptides using SAM (and a selenium-containing analogue) bearing an ethyl rather than methyl electrophile[11]. In a somewhat similar approach, Islam et al.[12] have previously developed a SAM co-factor analogue bearing a bulky alkynyl electrophilic group in place of the methyl group (Fig. 1b). Given the size of the electrophile, wild-type enzymes cannot utilise such a synthetic co-factor. It was

therefore necessary to engineer the target HKMT G9a, through a Y1154A mutation, which expands the size of the active site and allows for the transfer of the alkynyl group to G9a substrates. Subsequent enrichment and identification from cells is then possible via click chemistry. One disadvantage to this approach is the large alkynyl label which would likely interfere with downstream signalling reliant on the endogenous methyl marks and thus disrupt further evaluation of the chemical biology of HKMTs in a cellular context. It also remains unclear whether the Y1154A G9a/alkynyl SAM pair has an analogous substrate specificity profile to G9a, i.e., to what extent

substrates identified through this technique are *bone fide* G9a substrates. Nonetheless, this bump-and-hole approach allowed for selective substrate labelling for G9a using a bulky synthetic co-factor and engineered enzyme. We note that there are several other examples of the bump-and-hole approach applied to histone methyltransferases, through synthetic modification of the adenine or ribose parts of the SAM scaffold (Supplementary Table 1)[13,14]. As an alternative to bump-and-hole strategies, Luo and colleagues developed the non-natural co-factor, ProSeAM (Supplementary Table 1), which was shown to be compatible with several WT lysine and arginine methyltransferases. ProSeAM transfers a propargyl moiety, enabling pulldown/fluorescent tagging of substrates via Click chemistry[15]. ProSeAM was also subsequently utilised in a chemoproteomic approach developed by Sohtome et al., which expanded the scope of enzymes and substrates studied[16]. While these experiments demonstrated compatibility of a non-methyl co-factor with WT methyltransferases, the caveat remains that some native methyltransferases may not catalyse the transfer of a non-methyl label.

Herein we report proof of concept for an alternative methodology that combines the specificity of engineered co-factors with quantitative capabilities of isotopic labelling. We designed a heavy co-factor analogue, $^{13}CD_3$-BrSAM, that is highly selective for the HKMT DOT1L through a naturally occurring structural pocket ("hole") in the active site. As such, this approach allows the use of wild-type enzyme for selective heavy labelling of target substrates (Fig. 1c). Using $^{13}CD_3$-BrSAM, we isotopically label DOT1L substrates in vitro and in cellular lysates. We believe this approach should be translatable to the design of selective heavy co-factors for a range of additional HKMTs, further improving the biological understanding of this exciting class of enzymes.

## Results and discussion

### Development of a heavy co-factor analogue to study the activity of DOT1L

The design for our selective co-factor, $^{13}CD_3$-BrSAM, was based on the DOT1L inhibitor, bromo-deaza-SAH[17] (5, Fig. 2, Supplementary Fig. 1), which is an analogue of the endogenous by-product of protein methylation, S-adenosyl-L-homocysteine (SAH). Inhibitor 5 was previously reported to potently inhibit DOT1L (IC$_{50}$ = 77 nM), with high selectivity over other HKMTs ( > 2500 fold) and other methyltransferases ( > 20 fold). Importantly, 5 binds to the DOT1L active site without inducing significant remodelling of the conformation of the active site relative to SAH. Such a

binding mode contrasts with other reported DOT1L inhibitors, which induce significant structural change of the active site[18]. Upon inspection of the X-ray structure of 5 bound to DOT1L (Fig. 1d), we hypothesised that $^{13}CD_3$-BrSAM – the equivalent methyl donating sulfonium analogue – would be catalytically competent. This is due to the retained structure of the active site (SAH versus 5), coupled with the fact that the sulfur responsible for transfer of the methyl group is correctly positioned adjacent to the internal channel, where methyl transfer to the target lysine takes place[19]. Consequently, we hypothesised that turnover of $^{13}CD_3$-BrSAM by DOT1L would result in heavy labelling of DOT1L substrate(s), and this would be detectable by MS analysis. As such this would provide a novel methodology to track methylation of DOT1L, which is a therapeutically interesting target (see below). It is worth emphasising that in contrast to several HKMTs[2] that have numerous histone and non-histone substrates, DOT1L is reported to be highly selective for methylating H3K79[20-23]. At the outset of the study, only one other substrate of DOT1L, the androgen receptor (AR), had been identified[24]. Thus we aimed to develop an MS-based technology to confirm the selectivity of DOT1L and seek potential novel substrates.

The synthesis of $^{13}CD_3$-BrSAM (Fig. 2, see Chemical Synthesis section of the SI) was completed via adaptation of the original route to 5[17], followed by a heavy methylation step. Firstly, glycosylation of a benzoyl-protected ribose with 7-bromodeazaadenine 1 was facilitated by the silyl-Hilbert-Johnson reaction, which proceeded similarly to previous reports[25,26]. Subsequent global deprotection of the benzoyl groups and concomitant aromatic substitution of the chloride of the adenine ring was achieved by microwave irradiation (as an alternative to a pressure flask) in methanol saturated with ammonia. The resultant alcohol was converted to chloroadenosine 4 through reaction with thionyl chloride, substituting the previously used toxic HMPA with MeCN, and using NH$_4$OH as a milder alternative to NaOH base to quench. To promote the subsequent nucleophilic substitution of the chloride, the thiolactone form of L-homocysteine (L-HT) was first hydrolysed using NaOH, before treatment with 4 in the presence of potassium iodide, to afford bromo-deaza-SAH, 5 with overall improved yield over the prior method (44% vs 9%). The procedure for heavy methylation involved in situ formation of $^{13}CD_3$OTf from heavy methyl iodide and silver triflate, prior to treatment of 5 with the resultant solution, using conditions employed by Kuethe[27]. A non-selective control compound, $^{13}CD_3$-SAM (8) was also synthesised from SAH using this method. Note: diastereomers of 7 and 8 arising from the S-methylation steps were not resolved.

**Fig. 2 | Synthesis of $^{13}CD_3$-BrSAM, 7.** Bromo-deaza-SAH 5 was synthesized using modification of previously reported conditions[17] (see Supplementary Methods provided in the SI), then converted to 7 using a heavy methyl triflate reagent.

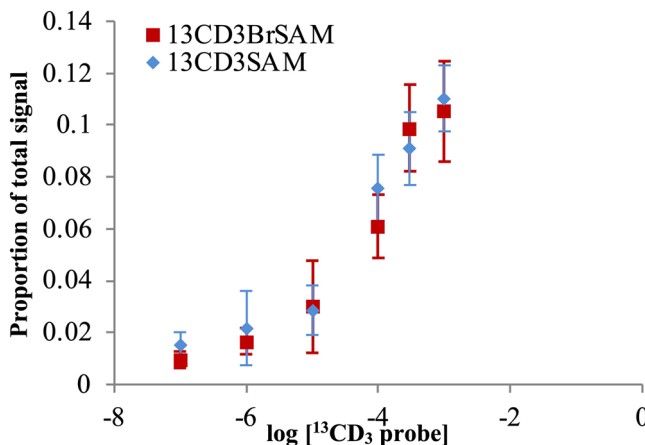

**Fig. 3 | Titration response curves for the treatment of nucleosomes with heavy labelled co-factor analogues in the presence of DOT1L.** The heavy mono-methyl peptide signal was quantified relative to the K79un-containing peptide standard (see also Supplementary Fig. 2 in the SI) and calculated as a proportion over the total unmodified, light mono-methyl and heavy mono-methyl signals, as carried out for the time course assays. Error = s.d. between three replicates. See Supplementary Methods provided in the SI.

## $^{13}CD_3$-BrSAM is catalytically competent and selective for DOT1L over other HKMTs

To test the compatibility of $^{13}CD_3$-BrSAM with DOT1L catalysis, in vitro assays were performed using commercially obtained nucleosomes extracted from HeLa cells and recombinant DOT1L, comparing the results to those obtained with $^{13}CD_3$-SAM. Quantitative kinetic studies demonstrated that the extent of heavy methylation observed following 15-min incubations of recombinant nucleosomes and DOT1L with each probe at various concentrations was very similar when utilizing $^{13}CD_3$-BrSAM as a co-factor versus $^{13}CD_3$-SAM (Fig. 3), confirming the catalytic competence of $^{13}CD_3$-BrSAM. Analysis of the histones revealed excellent incorporation of the heavy label at H3K79 – the target site of DOT1L–using either $^{13}CD_3$-SAM or $^{13}CD_3$-BrSAM as a co-factor, thus further confirming the functional competence of $^{13}CD_3$-BrSAM ($^{13}CD_3$-BrSAM labelling shown in Fig. 4a–c). No heavy labelling of the following histone lysine sites was observed by either co-factor when using DOT1L: H3K4, H3K9, H3K27, H3K36, H4K20, supporting the selectivity of DOT1L for H3K79.

We then assessed the selectivity of $^{13}CD_3$-BrSAM for DOT1L using a variety of representative alternative HMKTs. Specifically, we substituted DOT1L with the SET domain enzymes, SETD7, G9a, SETD2 and SUB420H2 which methylate H3K4, H3K9, H3K36 and H4K20 respectively. While treatment of nucleosomes using the heavy endogenous $^{13}CD_3$-SAM in the presence of a given methyltransferase enzyme resulted in the transfer of the heavy methyl group to their respective site, no such incorporation was observed under the same conditions using $^{13}CD_3$-BrSAM (Fig. 4d–g, Supplementary Figs. 3, 4, SI), confirming its excellent selectivity for DOT1L versus other HKMTs. Taken together, this data demonstrates that it is possible to develop a synthetic co-factor analogue for an individual HKMT, that is catalytically competent and highly selective for processing by that HKMT.

## Application of $^{13}CD_3$-BrSAM to cell lysates indicates the high specificity of DOT1L for H3K79

In mixed-lineage leukaemia (MLL), transposition of the MLL genes gives rise to fusion-proteins that aberrantly recruit DOT1L[28]. Dependency of DOT1L in MLL is therefore a promising drug target. To date, only a single substrate has been confidently identified for DOT1L (H3K79)[29], something we elected to further study using our methodology. Substrate discovery was performed using cytosolic extract from HEK293T cells that were pretreated with the DOT1L inhibitor EPZ5676 to increase DOT1L substrate

availability for heavy labelling[30]. DOT1L reactions were performed by adding recombinant DOT1L to cellular lysate along with SAM and $^{13}CD_3$-BrSAM at ratios of 1:9, 2:8, 3:7, 4:6 (1 mM). As an internal positive control, all conditions were supplemented with 2 μg of nucleosomes. Following the reaction with DOT1L, proteins were digested into peptides and fractionated for analysis by mass spectrometry. The presence of H3K79 heavy methylation was confirmed in all cases except the negative control (without DOT1L, Supplementary Fig. 5). Methylation of the androgen receptor (AR), which has also been reported as a DOT1L substrate[24], was not observed. We note however, that the methyl-containing peptide arising from tryptic digestion of AR would be 44 residues long, which is unfavourable for the MS conditions used in this study.

Initial screening for methyl-peptides of additional substrates was conducted using Mascot (Matrix Science), after removal of peptides identified from the negative control and manual screening for heavy/light pre-cursor mass pairs. From this search no peptides, other than H3K79, could be identified. We then used an in-house script developed in R that searches for the heavy/light mass pair (based on mass-shift and retention time) for peptides that have been identified in database searching as being methylated, but lack a peptide spectral match for the corresponding heavy/light label, similar to approaches that have been described elsewhere[31,32]. After removing heavy/light pairs that were visible in the negative control, no additional DOT1L substrates could be clearly identified. As a final search, we identified any pairs of MS2 spectra that had been acquired from a precursor with a 4 Da mass difference and eluted within 120 s of each other, regardless of a spectral match in either or both spectra. This subset of spectra was re-searched using Mascot and the spectra of identified heavy/light methyl pairs were then compared to identify matching peaks. From these matching peaks, we calculate the mass of the complementary fragment ion that would contain either the heavy or light methyl-modification, and identify these in their respective MS2 spectra, using HiLight-PTM[33], a tool we developed for this purpose. We then determined the Pearson's product moment correlation coefficient (R) between the matched fragment ions, removing spectra with R < 0.5 (Supplementary Fig. 6). From this analysis, three peptides were identified, corresponding to tRNA dimethylallyltransferase (dimethyl R408), K0825, (an uncharacterised protein, monomethyl R807) and His-tone H2B type W-T (H2BWT) (dimethyl K60). Only H2BWT had an internal methylated K/R, which is expected as tryptic peptidase activity is inhibited at methylated K/R residues. Inspection of the parent-ion peak areas show that the relative abundance of heavy label increases with the proportion of $^{13}CD_3$-BrSAM included in the reaction mix and the matched fragment ions had a high product moment correlation coefficient (R = 0.797, P < 0.001) (Supplementary Fig. 6, Supplementary Table 5 and 6). However, a peak corresponding to a heavy label was manually identified in the negative control. Given the relative abundance of this peak, it unlikely to be the result of activity from *endogenous* DOT1L, given our use of a DOT1L inhibitor in the sample preparation step. We therefore conclude that very little evidence was found for additional DOT1L substrates beyond H3K79.

## Conclusions

The discovery of novel locations for methylation in the proteome is challenging, owing to increased uncertainty about spectral identifications and challenges in accurately assigning the modified residue from MSMS data. Furthermore, the discovery of novel methylation sites is rarely linked to a specific catalytic enzyme, such that the biological context of methylation throughout the proteome is not well understood. The use of $^{13}CD_3$-BrSAM, a SAM analogue specific to DOT1L, addresses both challenges by providing enzyme-specific methyl-site analysis without the need to engineer or perturb the native enzyme while maintaining native (or near-native) reaction kinetics. We undertook discovery with $^{13}CD_3$-BrSAM in cell lysate to identify possible other substrates for DOT1L. Prior to this study, the literature was overwhelmingly supportive of a high selectivity of DOT1L for H3K79, with almost no other histone or non-histone substrates reported. We did not identify any conclusive evidence of other substrates of DOT1L. Given the high promiscuity of the other HKMTs[2], this is perhaps surprising,

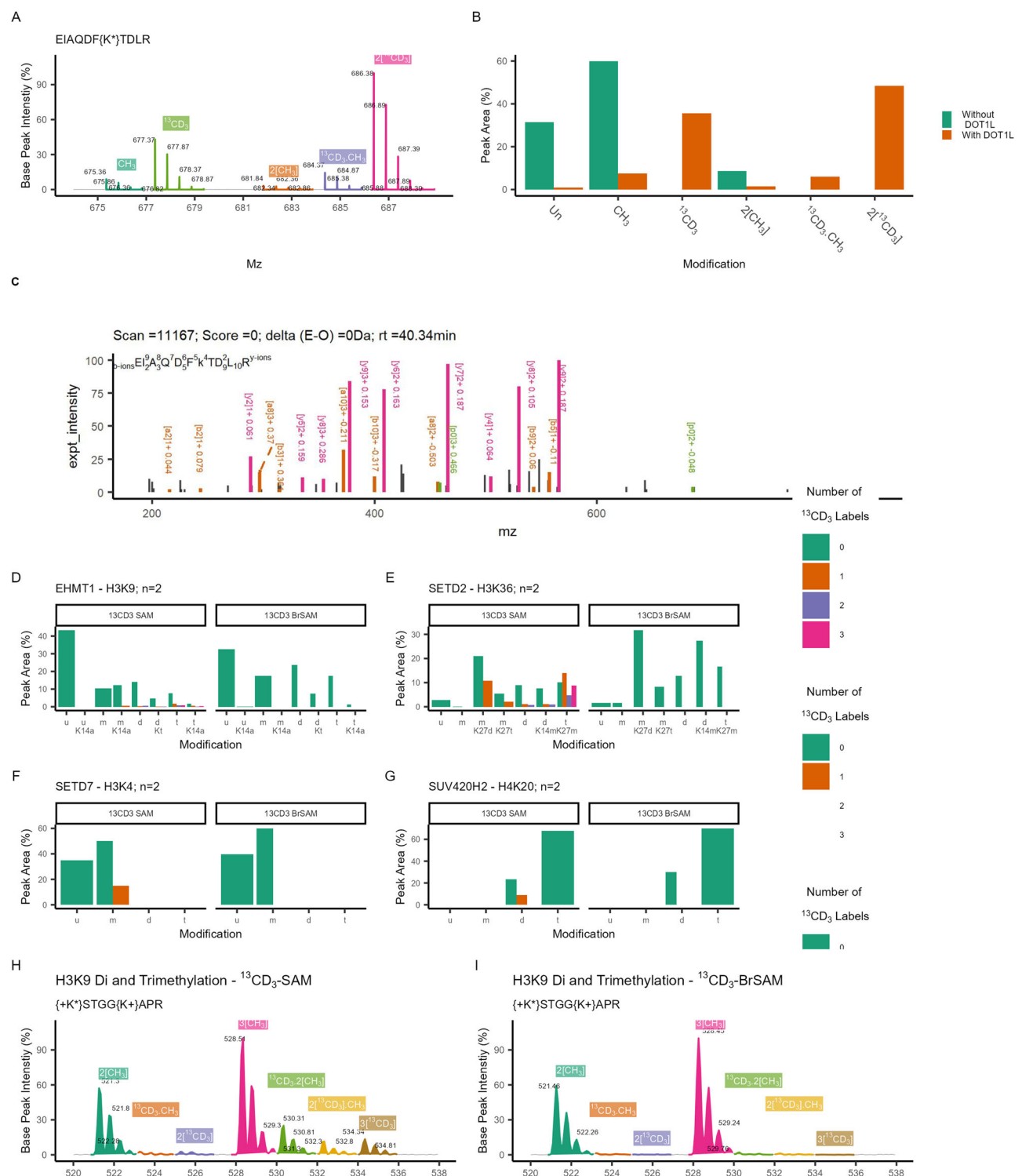

**Fig. 4 | MS detection of heavy-labelled nucleosomes incubated with methyl-transferases and heavy co-factors. a** Representative mass spectra of H3K79 showing masses corresponding to mono and di-methyl K79 with "light" ($CH_3$ – from endogenous/pre-existing methylation) and "heavy" ($^{13}CD_3$) labelling arising from transfer of isotopically labelled methyl-groups from $^{13}CD_3$-BrSAM. **b** The relative abundance (100 x abundance / sum of all abundances for each condition) of each H3K79 methyl peptide, calculated from the area of the extracted ion chromatograph for each peptidoforms. **c** An annotated fragmentation spectra for H3K79 heavy demethylation (2[$^{13}CD_3$]), orange = a/b-ion series, pink = y-ion series, green =

parent ion. The mass difference from the theoretical mass is given by each peak in Da (see Supplementary Table 4 for peak lists). rt = retention time. **d–g** Detection of heavy-methyl peaks H3K9, H3K36, H3K4, SUV420H2 after incubation with their respective modifying enzyme and heavy-labelled SAM ($^{13}CD_3$). There was no heavy labelling observed following the same reaction conditions in the presence of heavy labelled BrSAM. The number of heavy labels in the methyl-moiety is indicated (u = unmodified, m/d/t = mono/di/tri-methylation, a = acetylation). **h, i** Representative mass spectra showing heavy methyl-peaks observed for H3K9 in the presence of $^{13}CD_3$SAM but not $^{13}CD_3$-BrSAM.

but also supported by our results and others. Such selectivity may relate to the differences (non-SET domain) in the active site of DOT1L in comparison to other HKMTs[29]. While this study was focused on DOT1L, the concept reported herein should be translatable to other HKMTs by modelling of the SAM binding pocket and SAM-analogue configurations, providing a complementary method to improve biological understanding of these proteins.

## Methods

Details of all experimental procedures including biological assays are provided in the Supplementary Methods section of the SI. Supplementary Tables 2 and 3 detail the reagents used for nucleosome and cell lysate labelling experiments respectively. The synthesis of compounds **2-8** are provided in the Chemical Synthesis section of the SI.

### Reporting summary

Further information on research design is available in the Nature Portfolio Reporting Summary linked to this article.

### Data availability

Analytical spectra for **7** and **8** are provided as Supplementary Figs. 7–11 in the SI. Raw data is available upon request from the authors. The mass spectrometry proteomics data have been deposited to the ProteomeXchange Consortium via the PRIDE partner repository with the dataset identifier PXD052790.

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

## Acknowledgements

We would like to acknowledge Cancer Research UK (C33325/A236190) and the EPSRC for funding. M. J. F. would like to thank the EPSRC for an Established Career Fellowship (EP/R00188X/1). N. T. would like to acknowledge the Faculty of Natural Sciences at Imperial College London for a Schrodinger Scholarship. E. L. would like to acknowledge AstraZeneca for part funding her stipend through a case studentship. Support for H. J. W. and for further infrastructure was provided by the

National Institute for Health Research (NIHR) Imperial Biomedical Research Centre (BRC). H. J. W. would also like to acknowledge the Medical Research Council (MR.R02524X/1) and Cancer Research UK (EDDCPJT \100022) for funding. We would like to thank Professor Masoud Vedadi (University of Toronto) for donating the methyltransferase enzymes SETD7, EHMT1, SETD2, SUB420H2.

## Author contributions

M. J. F. and P. D. conceived the idea. N. T. optimized the chemistry, synthesized compounds and performed in vitro experiments with nucleosomes. Further assistance with synthesis was provided by E. L. and K. A. H. J. W. performed in vitro experiments with nucleosomes and cell lysates. H. J. W. and P. D. performed database searching and MS2 matching analysis. B. J. provided NMR spectra for compounds. N. T., H. J. W., and M. J. F. co-wrote the manuscript. All authors approved submission.

## Competing interests

The authors declare no competing interests.
