## [Peer Review File · Communications Chemistry]

Reviewers' comments:

Reviewer #1 (Remarks to the Author):

Trainor et al. present novel work on the synthesis and validation of a synthetic cofactor to profile histone lysine methylation by the histone methyltransferase enzyme DOT1L. The originality and novelty in this work lies in two factors, the design of the synthetic cofactor itself and the selectivity of the co-factor for DOT1L methyltransferase activity. The design is clever, the authors slightly modified a selective DOT1L inhibitor that does not induce significant changes in the conformation of DOT1L and modified this molecule into an agonist that retains a very similar activity profile to the deuterated version of the enzyme's native cofactor SAM (experiment fig. 3). Compared to other histone methyltransferase profiling strategies in previous literature with chemical probes/synthetic cofactors, the deuterated methylation strategy proposed by the authors is really a very subtle modification that should mimic very closely the native methylation mark on histone lysine residues installed by the DOT1L enzyme. This should be advantageous to previous strategies reported that incorporate quite significant chemical and biological modifications to these enzymes and their cofactors. The work in the mass spec proteomics clearly shows selective deuterated methylation only in the presence of DOT1L with the synthetic cofactor $^{13}\text{CD}_3$ BrSAM, this synthetic cofactor is not utilised by other histone methyl transferases (experiments in Fig 4D-G). When applying their synthetic cofactor to cell lysates it seems clear that the DOT1L enzyme is highly selective for H3K79, a finding that correlates with previous literature. Clearly, these synthetic cofactors are going to be valuable chemical probes/tools for the many researchers studying the effects of H3K79 methylation and its implications in disease. The strategy used could also potentially be applied to other histone methyl transferase enzymes. This is high quality work, and performed and written to a high standard and will be very well suited to the wide readership of Communications Chemistry. I congratulate the authors on their work and I would recommend for publication after the minor revisions suggested below.

It seems to me at least as the data is presented in Fig. 4, the synthetic cofactor ($^{13}\text{CD}_3$ BrSAM) does not catalyse tri-methylation ($3\times ^{13}\text{CD}_3$) with DOT1L? Was this expected? In general, there also seems to be less trimethylation with $^{13}\text{CD}_3$ -SAM with other histone methyl transferase enzymes. Could this be an artefact of the synthetic cofactor, maybe the deuterated atoms are slightly bigger in size and hinder tri-methylation? Or is this trend to be expected in general? The authors should comment on this observation.

'Firstly, glycosylation of a benzoyl-protected ribose with 7- bromodeazaadenine 1 was facilitated by the silyl-Hilbert-Johnson reaction, which proceeded similarly to previous reports.' Appropriate citations should be added here for anyone who wants to repeat this work and to credit the authors in the previous reports.

One of the reaction schemes in Figure 2 is a bit confusing. Compound 4 has an arrow going to compound 5, but then compound 4 is also listed as a reagent in step 2 of the same scheme. Its oddly presented and needs modifying. It may also be useful to add yields here as they are discussed in the main text.

For clarity, is the data shown in figure 4a-c produced in the presence of $^{13}\text{CD}_3$ BrSAM or $^{13}\text{CD}_3$ SAM? The main text says both, but I'm assuming this specific data representing these figures is generated from one or the other? I'm assuming $^{13}\text{CD}_3$ BrSAM as no $^{13}\text{CD}_3$ is incorporated without DOT1L in Fig 4b? Additionally for clarity, I assume the "light" methylation is coming from SAM present in the extracted nucleosomes, or previously methylated histone lysine residues by SAM before extraction? It would be helpful if these points are clarified.

I'm not sure how useful the ^{13}C spectrum data is in the supporting information for compounds 7 and 8, you can't see the carbon signals as the signal to noise ratio is poor. Is this for comparison with the HSQC data? if available, the raw HPLC or LCMS data with the retention times of these molecules would probably be more useful for anyone wanting to repeat the synthesis of these synthetic cofactors.

James T. Hodgkinson

Reviewer #2 (Remarks to the Author):

I thoroughly enjoyed reading this manuscript but would like to see a few minor revisions before being accepted.

- The title gives the impression that this technique can be widely used for all HMTs whereas it is only applicable to DOT1L—a little misleading.
- Apart from getting a ^{13}C spectrum of the $^{13}\text{CD}_3\text{-BrSAM}$, I am unsure of the reason for using ^{13}C as $\text{CD}_3\text{-BrSAM}$ would give the same result.
- Considering the ^{13}C enrichment, the ^{13}C spectra are disappointing as I would have expected a much better signal/noise ratio.
- How does the use of $^{13}\text{CD}_3\text{-BrSAM}$ affect the kinetics of methylation? A side-by-side comparison of the kinetics of $^{13}\text{CD}_3\text{-BrSAM}$ vs. SAM would be useful.
- There needs to be a discussion of the inhibition of DOT1L by BrSAH which is the by-product of methylation using $^{13}\text{CD}_3\text{-BrSAM}$. You are looking for novel DOT1L substrates, so while DOT1L is methylating H3K79 it is generating a potent inhibitor of DOT1L which is 8 times more potent than SAH (IC_{50} 77 vs 600 nM).
- If you are seeking novel substrates of DOT1L would the use of $^3\text{H-SAM}$ be a more sensitive method?

Happy to see discussion/rebuttal on above points.

I have included minor edits in the main manuscript and SI.

Reviewer 1

It seems to me at least as the data is presented in Fig. 4, the synthetic cofactor (13CD3 BrSAM) does not catalyse tri-methylation (3X 13CD3) with DOT1L? Was this expected? In general, there also seems to be less trimethylation with 13CD3-SAM with other histone methyl transferase enzymes. Could this be an artefact of the synthetic cofactor, maybe the deuterated atoms are slightly bigger in size and hinder tri-methylation? Or is this trend to be expected in general? The authors should comment on this observation.

This is a good observation and is correct; we do not observe H3K79 ¹³CD₃ trimethylation of DOT1L in the presence of ¹³CD₃-BrSAM. However, we also did not observe, and routinely do not see, H3K79 trimethylation with the *in vitro* reaction performed on purified nucleosomes regardless of the methyl-donor. Methylation of H3K79 occurs sequentially and the proportion of endogenous di-methyl H3K79 on the purified nucleosomes is relatively low. Therefore there is limited di-methyl precursor to further methylate at the start of the *in vitro* reaction (Fig 4b). Given this, we do not believe the lack of trimethylation is an artifact of the isotopic labelling, or that if we increased the reaction time we would see trimethylated H3K79. It is perhaps also relevant that we did not check for the presence of H2B K120 mono-ubiquitination is known to be important for the generation of H3K79 trimethylation.

'Firstly, glycosylation of a benzoyl-protected ribose with 7-bromodeazaadenine 1 was facilitated by the silyl-Hilbert-Johnson reaction, which proceeded similarly to previous reports.' Appropriate citations should be added here for anyone who wants to repeat this work and to credit the authors in the previous reports.

We thank the reviewer for this comment and have added primary references to support readers in reproducing the work.

One of the reaction schemes in Figure 2 is a bit confusing. Compound 4 has an arrow going to compound 5, but then compound 4 is also listed as a reagent in step 2 of the same scheme. Its oddly presented and needs modifying. It may also be useful to add yields here as they are discussed in the main text.

Figure 2 has now been updated for clarification and to include yields as requested.

For clarity, is the data shown in figure 4a-c produced in the presence of 13CD3 BrSAM or 13CD3 SAM? The main text says both, but I'm assuming this specific data representing these figures is generated from one or the other? I'm assuming 13CD3 BrSAM as no 13CD3 is incorporated without DOT1L in Fig 4b? Additionally for clarity, I assume the "light" methylation is coming from SAM present in the extracted nucleosomes, or previously methylated histone lysine residues by SAM before extraction? It would be helpful if these points are clarified.

This is correct, heavy labelling in Fig 4 A-C was generated in the presence of 13CD3-BrSAM. The "light" signal arises from endogenous/pre-existing methylation on the peptides. The text and figure legend are modified as below to clarify these points:

Figure legend: (a) Representative mass spectra of H3K79 showing masses corresponding to mono and di-methyl K79 with "light" (CH₃ – from endogenous/pre-existing methylation) and "heavy" (¹³CD₃) labelling arising from transfer of isotopically labelled methyl-groups from ¹³CD₃-BrSAM.

Text: Analysis of the histones revealed excellent incorporation of the heavy label at H3K79 – the target site of DOT1L - using either $^{13}\text{CD}_3$ -SAM or $^{13}\text{CD}_3$ -BrSAM as a co-factor, thus further confirming the functional competence of $^{13}\text{CD}_3$ -BrSAM ($^{13}\text{CD}_3$ -BrSAM labelling shown in Fig. 4a-c).

I'm not sure how useful the ^{13}C spectrum data is in the supporting information for compounds 7 and 8, you can't see the carbon signals as the signal to noise ratio is poor. Is this for comparison with the HSQC data? if available, the raw HPLC or LCMS data with the retention times of these molecules would probably be more useful for anyone wanting to repeat the synthesis of these synthetic cofactors.

We thank the reviewer for this feedback, which is consistent with feedback from Reviewer 2. In our study, we only routinely made low (mg) quantities of **7** and **8** and did not initially collect ^{13}C NMR data. We attempted to generate ^{13}C NMR data for this submission, but the low quantity (recovered from biological aliquots) of samples meant that we were unable to generate high quality data, unfortunately. We note that we were able to improve the signal to noise for the ^1H spectrum of $^{13}\text{CD}_3$ -BrSAM by using a higher frequency instrument, a thinner sample tube, and a water suppression method for the acquisition. Unfortunately, these measures did not afford the same improvement in the acquisition of ^{13}C spectra. As such, we have removed the ^{13}C spectra from the Supporting Information, as it may be misleading due to the signal-to-noise ratio. However, we have retained the HSQC data, which is useful for assigning the ^{13}C signals. At the request of the reviewer, we have also provided the retention times of these compounds under the standard HPLC method reported.

Reviewer 2

The title gives the impression that this technique can be widely used for all HMTs whereas it is only applicable to DOT1L—a little misleading.

We thank the reviewer for this feedback. We have amended the title in response to this point to focus on DOT1L.

Apart from getting a ^{13}C spectrum of the $^{13}\text{CD}_3$ -BrSAM, I am unsure of the reason for using ^{13}C as $^{13}\text{CD}_3$ -BrSAM would give the same result.

The use of ^{13}C CD_3 gives an additional mass shift over CD_3 . This is useful for the MS proteomics to further separate peaks of interest from artefactual peaks.

Considering the ^{13}C enrichment, the ^{13}C spectra are disappointing as I would have expected a much better signal/noise ratio.

Please see the response to reviewer 1 about the ^{13}C NMR data.

How does the use of $^{13}\text{CD}_3$ -BrSAM affect the kinetics of methylation? A side-by-side comparison of the kinetics of $^{13}\text{CD}_3$ -BrSAM vs. SAM would be useful.

In Figure 3 we show kinetic data for $^{13}\text{CD}_3$ -SAM vs $^{13}\text{CD}_3$ -BrSAM, which enables us to infer that no significant difference exists between the activity of these two co-factors for DOT1L. We believe the reviewer may be referring to the difference between using a co-factor with a ^{13}C heavy label ($^{13}\text{CD}_3$) versus a ^{12}C heavy label (CD_3). Unfortunately, we did not perform such an experiment since we did not focus on a ^{12}C heavy label (CD_3) for the reasons given above. While this would be an interesting comparison for a future study, we do not believe changes in ^{12}C versus ^{13}C kinetics would change the outcome of our study.

There needs to be a discussion of the inhibition of DOT1L by BrSAH which is the by-product of methylation using $^{13}\text{CD}_3\text{-BrSAM}$. You are looking for novel DOT1L substrates, so while DOT1L is methylating H3K79 it is generating a potent inhibitor of DOT1L which is 8 times more potent than SAH (IC₅₀ 77 vs 600 nM).

This is an insightful comment from the reviewer. In methylation reactions, we add exogenous DOT1L and an excess of SAM/BrSAM such that over the course of the 2 hour reaction, these are in excess of their inhibitory forms (SAH/BrSAH). In Figure 3, we show that in similar conditions, the kinetics of $^{13}\text{CD}_3\text{-SAM}$ and $^{13}\text{CD}_3\text{-BrSAM}$ are the same.

If you are seeking novel substrates of DOT1L would the use of $^3\text{H-SAM}$ be a more sensitive method?

The use of $^3\text{H-SAM}$ would allow for global methylation labelling, but would make discrimination of methylation arising from other endogenous methyltransferases more challenging. We still believe the advantage of using methyl-transferase specific co-factors is to maintain the link between the methylated protein substrate and the responsible enzyme and with the additional benefit of the absence of non-DOT1L mediated methylation background, providing better signal: noise. While a $^3\text{H-BrSAM}$ probe would potentially be interesting, we believe the $^{13}\text{CD}_3$ label serves as a more convenient and stable isotope label for synthesis and handling, since the safety requirements of handling radioisotopes are not required.

REVIEWERS' COMMENTS:

Reviewer #1 (Remarks to the Author):

The authors have addressed all my queries/edits in the revised manuscript, congratulations.

Reviewer #2 (Remarks to the Author):

The authors have answered all my questions concerning this manuscript including editing the title. The manuscript is now acceptable for publication.